# ACTIVE NEURAL LOCALIZATION

**Devendra Singh Chaplot, Emilio Parisotto, Ruslan Salakhutdinov**
Machine Learning Department
School of Computer Science
Carnegie Mellon University
{chaplot,eparisot,rsalakhu}@cs.cmu.edu

## ABSTRACT

Localization is the problem of estimating the location of an autonomous agent from an observation and a map of the environment. Traditional methods of localization, which filter the belief based on the observations, are sub-optimal in the number of steps required, as they do not decide the actions taken by the agent. We propose "Active Neural Localizer", a fully differentiable neural network that learns to localize accurately and efficiently. The proposed model incorporates ideas of traditional filtering-based localization methods, by using a structured belief of the state with multiplicative interactions to propagate belief, and combines it with a policy model to localize accurately while minimizing the number of steps required for localization. Active Neural Localizer is trained end-to-end with reinforcement learning. We use a variety of simulation environments for our experiments which include random 2D mazes, random mazes in the Doom game engine and a photo-realistic environment in the Unreal game engine. The results on the 2D environments show the effectiveness of the learned policy in an idealistic setting while results on the 3D environments demonstrate the model's capability of learning the policy and perceptual model jointly from raw-pixel based RGB observations. We also show that a model trained on random textures in the Doom environment generalizes well to a photo-realistic office space environment in the Unreal engine.

## 1 INTRODUCTION

Localization is the problem of estimating the position of an autonomous agent given a map of the environment and agent observations. The ability to localize under uncertainty is required by autonomous agents to perform various downstream tasks such as planning, exploration and target-navigation. Localization is considered as one of the most fundamental problems in mobile robotics (Cox & Wilfong, 1990; Borenstein et al., 1996). Localization is useful in many real-world applications such as autonomous vehicles, factory robots and delivery drones.

In this paper we tackle the global localization problem where the initial position of the agent is unknown. Despite the long history of research, global localization is still an open problem, and there are not many methods developed which can be learnt from data in an end-to-end manner, instead typically requiring significant hand-tuning and feature selection by domain experts. Another limitation of majority of localization approaches till date is that they are *passive*, meaning that they passively estimate the position of the agent from the stream of incoming observations, and do not have the ability to decide the actions taken by the agent. The ability to decide the actions can result in faster as well as more accurate localization as the agent can learn to navigate quickly to unambiguous locations in the environment.

We propose "Active Neural Localizer", a neural network model capable of *active* localization using raw pixel-based observations and a map of the environment[1][2]. Based on the Bayesian filtering algorithm for localization (Fox et al., 2003), the proposed model contains a perceptual model to estimate the likelihood of the agent's observations, a structured component for representing the belief, multiplicative interactions to propagate the belief based on observations and a policy model over the current belief to localize accurately while minimizing the number of steps required for localization. The entire model is fully differentiable and trained using reinforcement learning, allowing the perceptual model and the policy model to be learnt simultaneously in an end-to-end fashion. A variety

---

[1] Demo videos: https://devendrachaplot.github.io/projects/Neural-Localization

[2] The code is available at https://github.com/devendrachaplot/Neural-Localization

of 2D and 3D simulation environments are used for testing the proposed model. We show that the Active Neural Localizer is capable of generalizing to not only unseen maps in the same domain but also across domains.

## 2 RELATED WORK

Localization has been an active field of research since more than two decades. In the context of mobile autonomous agents, Localization can be refer to two broad classes of problems: Local localization and Global localization. Local localization methods assume that the initial position of the agent is known and they aim to track the position as it moves. A large number of localization methods tackle only the problem of local localization. These include classical methods based on Kalman Filters (Kalman et al., 1960; Smith et al., 1990) geometry-based visual odometry methods (Nistér et al., 2006) and most recently, learning-based visual odometry methods which learn to predict motion between consecutive frames using recurrent convolutional neural networks (Clark et al., 2017b; Wang et al., 2017). Local localization techniques often make restrictive assumptions about the agent's location. Kalman filters assume Gaussian distributed initial uncertainty, while the visual odometry-based methods only predict the relative motion between consecutive frames or with respect to the initial frame using camera images. Consequently, they are unable to tackle the global localization problem where the initial position of the agent is unknown. This also results in their inability to handle localization failures, which consequently leads to the requirement of constant human monitoring and intervention (Burgard et al., 1998).

Global localization is more challenging than the local localization problem and is also considered as the basic precondition for truly autonomous agents by Burgard et al. (1998). Among the methods for global localization, the proposed method is closest to Markov Localization (Fox, 1998). In contrast to local localization approaches, Markov Localization computes a probability distribution over all the possible locations in the environment. The probability distribution also known as the *belief* is represented using piecewise constant functions (or histograms) over the state space and propagated using the Markov assumption. Other methods for global localization include Multi-hypothesis Kalman filters (Cox & Leonard, 1994; Roumeliotis & Bekey, 2000) which use a mixture of Gaussians to represent the belief and Monte Carlo Localization (Thrun et al., 2001) which use a set of samples (or *particles*) to represent the belief.

All the above localization methods are *passive*, meaning that they aren't capable of deciding the actions to localize more accurately and efficiently. There has been very little research on *active* localization approaches. Active Markov Localization (Fox et al., 1998) is the active variant of Markov Localization where the agent chooses actions greedily to maximize the reduction in the entropy of the belief. Jensfelt & Kristensen (2001) presented the active variant of Multi-hypothesis Kalman filters where actions are chosen to optimise the information gathering for localization. Both of these methods do not learn from data and have very high computational complexity. In contrast, we demonstrate that the proposed method is several order of magnitudes faster while being more accurate and is capable of learning from data and generalizing well to unseen environments.

Recent work has also made progress towards end-to-end localization using deep learning models. Mirowski et al. (2016) showed that a stacked LSTM can do reasonably well at self-localization. The model consisted of a deep convolutional network which took in at each time step state observations, reward, agent velocity and previous actions. To improve performance, the model also used several auxiliary objectives such as depth prediction and loop closure detection. The agent was successful at navigation tasks within complex 3D mazes. Additionally, the hidden states learned by the models were shown to be quite accurate at predicting agent position, even though the LSTM was not explicitly trained to do so. Other works have looked at doing end-to-end relocalization more explicitly. One such method, called PoseNet (Kendall et al., 2015), used a deep convolutional network to implicitly represent the scene, mapping a single monocular image to a 3D pose (position and orientation). This method is limited by the fact that it requires a new PoseNet trained on each scene since the map is represented implicitly by the convnet weights, and is unable to transfer to scenes not observed during training. An extension to PoseNet, called VidLoc (Clark et al., 2017a), utilized temporal information to make more accurate estimates of the poses by passing a Bidirectional LSTM over each monocular image in a sequence, enabling a trainable smoothing filter over the pose estimates. Both these methods lack a straightforward method to utilize past map data to do localization in a new environment. In contrast, we demonstrate our method is capable of generalizing to new maps that were not previously seen during training time.

## 3 Background: Bayesian Filtering

Bayesian filters (Fox et al., 2003) are used to probabilistically estimate a dynamic system's state using observations from the environment and actions taken by the agent. Let $y_t$ be the random variable representing the state at time $t$. Let $s_t$ be the observation received by the agent and $a_t$ be the action taken by the agent at time step $t$. At any point in time, the probability distribution over $y_t$ conditioned over past observations $s_{1:t-1}$ and actions $a_{1:t-1}$ is called the *belief*, $Bel(y_t) = p(y_t|s_{1:t-1}, a_{1:t-1})$ The goal of Bayesian filtering is to estimate the belief sequentially. For the task of localization, $y_t$ represents the location of the agent, although in general it can represent the state of the any object(s) in the environment. Under the Markov assumption, the belief can be recursively computed using the following equations:

$$\overline{Bel}(y_t) = \sum_{y_{t-1}} p(y_t|y_{t-1}, a_{t-1})Bel(y_{t-1}), \quad Bel(y_t) = \frac{1}{Z}Lik(s_t)\overline{Bel}(y_t),$$

where $Lik(s_t) = p(s_t|y_t)$ is the likelihood of observing $s_t$ given the location of the agent is $y_t$, and $Z = \Sigma_{y_t} Lik(s_t)\overline{Bel}(y_t)$ is the normalization constant. The likelihood of the observation, $Lik(s_t)$ is given by the *perceptual model* and $p(y_t|y_{t-1}, a_{t-1})$, i.e. the probability of landing in a state $y_t$ from $y_{t-1}$ based on the action, $a_{t-1}$, taken by the agent is specified by a state transition function, $f_t$. The belief at time $t = 0$, $Bel(y_0)$, also known as the *prior*, can be specified based on prior knowledge about the location of the agent. For global localization, prior belief is typically uniform over all possible locations of the agent as the agent position is completely unknown.

## 4 Methods

### 4.1 Problem Formulation

Let $s_t$ be the observation received by the agent and $a_t$ be the action taken by the agent at time step $t$. Let $y_t$ be a random variable denoting the state of the agent, that includes its x-coordinate, y-coordinate and orientation. In addition to agent's past observations and actions, a localization algorithm requires some information about the map, such as the map design. Let the information about the map be denoted by $M$. In the problem of active localization, we have two goals: (1) Similar to the standard state estimation problem in the Bayesian filter framework, the goal is to estimate the *belief*, $Bel(y_t)$, or the probability distribution of $y_t$ conditioned over past observations and actions and the information about the map, $Bel(y_t) = p(y_t|s_{1:t}, a_{1:t-1}, M)$, (2) We also aim to learn a policy $\pi(a_t|Bel(y_t))$ for localizing accurately and efficiently.

### 4.2 Proposed Model

**Representation of Belief and Likelihood**   Let $y_t$ be a tuple $A_o, A_x, A_y$ where $A_x, A_y$ and $A_o$ denote agent's x-coordinate, y-coordinate and orientation respectively. Let $M \times N$ be the map size, and $O$ be the number of possible orientations of the agent. Then, $A_x \in [1, M]$, $A_y \in [1, N]$ and $A_o \in [1, O]$. Belief is represented as an $O \times M \times N$ tensor, where $(i, j, k)^{th}$ element denotes the belief of agent being in the corresponding state, $Bel(y_t = i, j, k)$. This kind of grid-based representation of belief is popular among localization methods as if offers several advantages over topological representations (Burgard et al., 1996; Fox et al., 2003). Let $Lik(s_t) = p(s_t|y_t)$ be the likelihood of observing $s_t$ given the location of the agent is $y_t$, The likelihood of an observation in a certain state is also represented by an $O \times M \times N$ tensor, where $(i, j, k)^{th}$ element denotes the likelihood of the current observation, $s_t$ given that the agent's state is $y_t = i, j, k$. We refer to these tensors as Belief Map and Likelihood Map in the rest of the paper.

**Model Architecture**   The overall architecture of the proposed model, Active Neural Localizer (ANL), is shown in Figure 1. It has two main components: the perceptual model and the policy model. At each timestep $t$, the *perceptual model* takes in the agent's observation, $s_t$ and outputs the Likelihood Map $Lik(s_t)$. The belief is propagated through time by taking an element-wise dot product with the Likelihood Map at each timestep. Let $\overline{Bel}(y_t)$ be the Belief Map at time $t$ before observing $s_t$. Then the belief, after observing $s_t$, denoted by $Bel(y_t)$, is calculated as follows:

$$Bel(y_t) = \frac{1}{Z}\overline{Bel}(y_t) \odot Lik(s_t)$$

where $\odot$ denotes the Hadamard product, $Z = \sum_{y_t} Lik(s_t)\overline{Bel}(y_t)$ is the normalization constant.

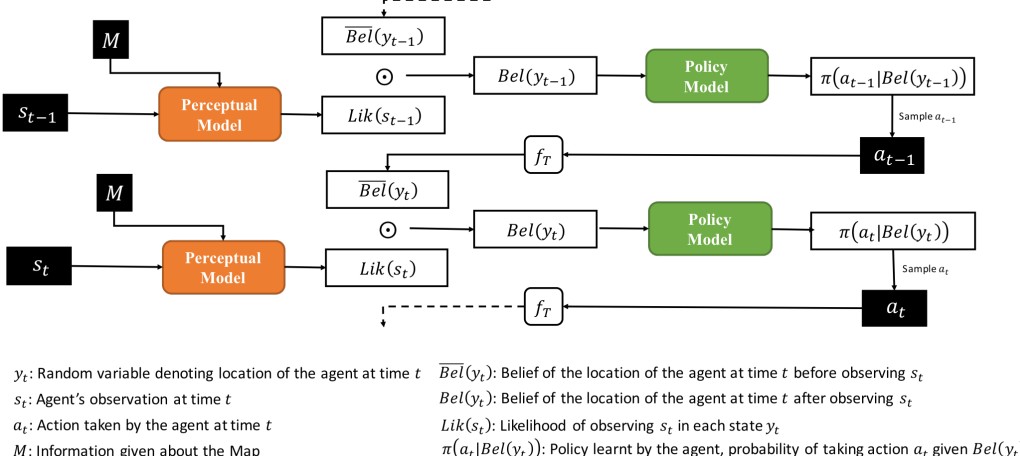

$y_t$: Random variable denoting location of the agent at time $t$    $\overline{Bel}(y_t)$: Belief of the location of the agent at time $t$ before observing $s_t$

$s_t$: Agent's observation at time $t$    $Bel(y_t)$: Belief of the location of the agent at time $t$ after observing $s_t$

$a_t$: Action taken by the agent at time $t$    $Lik(s_t)$: Likelihood of observing $s_t$ in each state $y_t$

$M$: Information given about the Map    $\pi(a_t|Bel(y_t))$: Policy learnt by the agent, probability of taking action $a_t$ given $Bel(y_t)$

$\odot$: Element-wise dot product    $f_T$: Transition function

**Figure 1:** The architecture of the proposed model. The perceptual model computes the likelihood of the current observation in all possible locations. The belief of agent's location is propagated through time by taking element-wise dot-product with the likelihood. The policy model learns a policy to localize accurately while minimizing the number of steps required for localization. See text for more details.

The Belief Map, after observing $s_t$, is passed through the *policy model* to obtain the probability of taking any action, $\pi(a_t|Bel(y_t))$. The agent takes an action $a_t$ sampled from this policy. The Belief Map at time $t+1$ is calculated using the transition function ($f_T$), which updates the belief at each location according to the action taken by the agent, i.e. $p(y_{t+1}|y_t, a_t)$. The transition function is similar to the egomotion model used by Gupta et al. (2017) for mapping and planning. For 'turn left' and 'turn right' actions, the transition function just swaps the belief in each orientation. For the the 'move forward' action, the belief values are shifted one unit according to the orientation. If the next unit is an obstacle, then the value doesn't shift, indicating a collison (See Appendix B for more details).

### 4.3 MODEL COMPONENTS

**Perceptual Model**    The perceptual model computes the feature representation from the agent's observation and the states given in the map information. The likelihood of each state in the map information is calculated by taking the cosine similarity of the feature representation of the agent's observation with the feature representation of the state. Cosine similarity is commonly used for computing the similarity of representations (Nair & Hinton, 2010; Huang et al., 2013) and has also been used in the context on localization (Chaplot et al., 2016). The benefits of using cosine similarity over dot-product have been highlighted by Chunjie et al. (2017).

In the 2D environments, the observation is used to compute a one-hot vector of the same dimension representing the depth which is used as the feature representation directly. This resultant Likelihood map has uniform non-zero probabilities for all locations having the observed depth and zero probabilities everywhere else. For the 3D environments, the feature representation of each observation is obtained using a trainable deep convolutional network (LeCun et al., 1995) (See Appendix B for architecture details). Figure 2 shows examples of the agent observation and the corresponding Likelihood Map computed in both 2D and 3D environments. The simulation environments are described in detail in Section 5.1.

**Policy Model**    The policy model gives the probablity of the next action given the current belief of the agent. It is trained using reinforcement learning, specifically Asynchronous Advantage Actor-Critic (A3C) (Mnih et al., 2016) algorithm (See Appendix A for a brief background on reinforcement learning). The belief map is stacked with the map design matrix and passed through 2 convolutional layers followed by a fully-connected layer to predict the policy as well as the value function. The policy and value losses are computed using the rewards observed by the agent and backpropagated through the entire model (See Appendix B for architecture details).

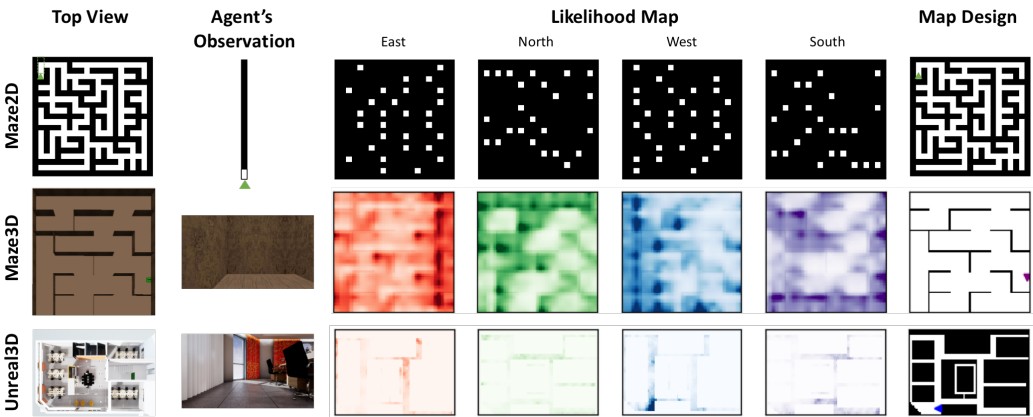

**Figure 2:** The map design, agent's observation and the corresponding likelihood maps in different domains. In 2D domains, agent's observation is the pixels in front of the agent until the first obstacle. In the 3D domain, the agent's observation is the image showing the first-person view of the world as seen by the agent.

## 5 EXPERIMENTS

As described in Section 4, agent's state, $y_t$ is a tuple $A_o, A_x, A_y$ where $A_x, A_y$ and $A_o$ denote agent's x-coordinate, y-coordinate and orientation respectively. $A_x \in [1, M]$, $A_y \in [1, N]$ and $A_o \in [1, O]$, where $M \times N$ is the map size, and $O$ be the number of possible orientations of the agent. We use a variety of domains for our experiments. The values of $M$ and $N$ vary accross domains but $O = 4$ is fixed. The possible actions in all domains are 'move forward', 'turn left' and 'turn right'. The turn angle is fixed at $(360/O = 90)$. This ensures that the agent is always in one of the 4 orientations, North, South, East and West. Note that although we use, $O = 4$ in all our experiments, our method is defined for any value of $O$. At each time step, the agent receives an intermediate reward equal to the maximum probability of being in any state, $r_t = max_{y_t}(Bel(y_t))$. This encourages the agent the reduce the entropy of the Belief Map in order to localize as fast as possible. We observed that the agent converges to similar performance without introducing the intermediate reward, but it helps in speeding up training. At the end of the episode, the prediction is the state with the highest probability in the Belief Map. If the prediction is correct, i.e. $y^* = \arg\max_{y_t} Bel(y_t)$ where $y*$ is the true state of the agent, then the agent receives a positive reward of $1$. Please refer to Appendix B for more details about training and hyper-parameters. The metric for evaluation is accuracy (Acc) which refers to the ratio of the episodes where the agent's prediction was correct over 1000 episodes. We also report the total runtime of the method in seconds taken to evaluate 1000 episodes.

### 5.1 SIMULATION ENVIRONMENTS

**Maze 2D**    In the Maze2D environment, maps are represented by a binary matrix, where 0 denotes an obstacle and 1 denotes free space. The map designs are generated randomly using Kruskal's algorithm Kruskal (1956). The agent's observation in 2D environments is the series of pixels in front of the agent. For a map size of $M \times N$, the agent's observation is an array of size $max(M, N)$ containing pixels values in front of the agent. The view of the agent is obscured by obstacles, so all pixel values behind the first obstacle are treated as 0. The information about the map, $M$, received by the agent is the matrix representing the map design. Note that the observation at any state in the map can be computed using the map design. The top row in Figure 2 shows examples of map design and agent's observation in this environment.

The 2D environments provide ideal conditions for Bayesian filtering due to lack of observation or motion noise. The experiments in the 2D environments are designed to evaluate and quantify the effectiveness of the policy learning model in ideal conditions. The size of the 2D environments can also be varied to test the scalability of the policy model. This design is similar to previous experimental settings such as by Tamar et al. (2016) and Karkus et al. (2017) for learning a target-driven navigation policy in grid-based 2D environments.

**3D Environments**    In the 3D environments, the observation is an RGB image of the first-person view of the world as seen by the agent. The x-y coordinates of the agent are continuous variables,

**Table 1:** Results on the 2D Environments. 'Time' refers to the number of seconds required to evaluate 1000 episodes with the corresponding method and 'Acc' stands for accuracy over 1000 episodes.

| Env | | Maze2D | | | | | | All |
|---|---|---|---|---|---|---|---|---|
| Map Size | | 7x7 | | 15x15 | | 21x21 | | |
| Episode Length | | 15 | 30 | 20 | 40 | 30 | 60 | |
| Markov | Time | 12 | 15 | 29 | 31 | 49 | 51 | 31.2 |
| Localization | Acc | 0.334 | 0.529 | 0.351 | 0.606 | 0.414 | 0.661 | 0.483 |
| Active Markov | Time | 29 | 53 | 72 | 165 | 159 | 303 | 130.2 |
| Localization (Fast) | Acc | 0.436 | 0.619 | 0.468 | 0.657 | 0.512 | 0.735 | 0.571 |
| Active Markov | Time | 1698 | 3066 | 3791 | 8649 | 8409 | 13554 | 6527.8 |
| Localization (Slow) | Acc | 0.854 | 0.938 | 0.846 | **0.984** | 0.845 | 0.958 | 0.904 |
| Active Neural | Time | 22 | 34 | 44 | 66 | 82 | 124 | 62.0 |
| Localization | Acc | **0.936** | **0.939** | **0.905** | 0.939 | **0.899** | **0.984** | **0.934** |

unlike the discrete grid-based coordinates in the 2D environments. The matrix denoting the belief of the agent is discretized meaning each pixel in the Belief map corresponds to a range of states in the environment. At the start of every epsiode, the agent is spawned at a random location in this continuous range as opposed to a discrete location corresponding to a pixel in the belief map for 2D environments. This makes localization much more challenging than the 2D envrionments. Apart from the map design, the agent also receives a set of images of the visuals seen by the agent at a few locations uniformly placed around the map in all 4 orientations. These images, called memory images, are required by the agent for global localization. They are not very expensive to obtain in real-world environments. We use two types of 3D environments:

**Maze3D:** Maze3D consists of virtual 3D maps built using the Doom Game Engine. We use the ViZDoom API (Kempka et al., 2016) to interact with the gane engine. The map designs are generated using Kruskal's algorithm and Doom maps are created based on these map designs using Action Code Scripts[3]. The design of the map is identical to the Maze2D map designs with the difference that the paths are much thicker than the walls as shown in Figure 2. The texture of each wall, floor and ceiling can be controlled which allows us to create maps with different number of 'landmarks'. Landmarks are defined to be walls with a unique texture.

**Unreal3D:** Unreal3D is a photo-realistic simulation environment built using the Unreal Game Engine. We use the AirSim API (Shah et al., 2017) to interact with the game engine. The environment consists of a modern office space as shown in Figure 2 obtained from the Unreal Engine Marketplace[4].

The 3D environments are designed to test the ability of the proposed model to jointly learn the perceptual model along with the policy model as the agent needs to handle raw pixel based input while learning a policy. The Doom environment provides a way to test the model in challenging ambiguous environments by controlling the number of landmarks in the environment, while the Unreal Environment allows us to evaluate the effectiveness of the model in comparatively more realistic settings.

## 5.2 BASELINES

**Markov Localization** (Fox, 1998) is a passive probabilistic approach based on Bayesian filtering. We use a geometric variant of Markov localization where the state space is represented by fine-grained, regularly spaced grid, called position probability grids (Burgard et al., 1996), similar to the state space in the proposed model. Grid-based state space representations is known to offer several advantages over topological representations (Burgard et al., 1996; Fox et al., 2003). In the passive localization approaches actions taken by the agent are random.

**Active Markov Localization** (AML) (Fox et al., 1998) is the active variant of Markov Localization where the actions taken by the agent are chosen to maximize the ratio of the 'utility' of the action to the 'cost' of the action. The 'utility' of an action $a$ at time $t$ is defined as the expected reduction in the uncertainity of the agent state after taking the action $a$ at time $t$ and making the next observation

---

[3] https://en.wikipedia.org/wiki/Action_Code_Script
[4] https://www.unrealengine.com/marketplace/small-office-prop-pack

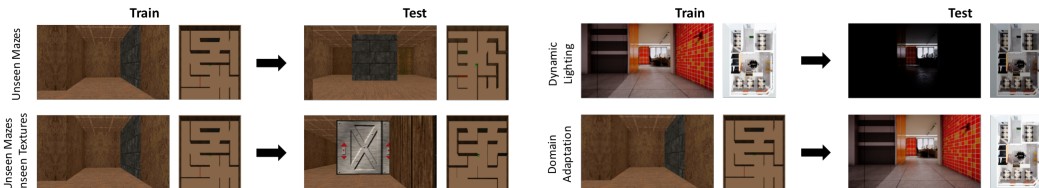

**Figure 3:** Different experiments in the 3D Environments. Refer to the text for more details.

at time $t + 1$: $U_t(a) = H(y_t) - \mathbb{E}_a[H(y_{t+1})]$, where $H(y)$ denotes the entropy of the belief: $H(y) = -\sum_y Bel(y) \log Bel(y)$, and $\mathbb{E}_a[H(y_{t+1})]$ denotes the expected entropy of the agent after taking the action $a$ and observing $y_{t+1}$. The 'cost' of an action refers to the time needed to perform the action. In our environment, each action takes a single time step, thus the cost is constant.

We define a generalized version of the AML algorithm. The utility can be maximized over a sequence of actions rather than just a single action. Let $a^* \in \mathbb{A}^{n_l}$ be the action sequence of length $n_l$ that maximizes the utility at time $t$, $a^* = \arg\max_a U_t(a)$ (where $\mathbb{A}$ denotes the action space). After computing $a^*$, the agent need not take all the actions in $a^*$ before maximizing the utility again. This is because new observations made while taking the actions in $a^*$ might affect the utility of remaining actions. Let $n_g \in \{1, 2, ..., n_l\}$ be the number of actions taken by the agent, denoting the greediness of the algorithm. Due to the high computational complexity of calculating utility, the agent performs random action until belief is concentrated on $n_m$ states (ignoring beliefs under a certain threshold). The complexity of the generalized AML is $\mathcal{O}(n_m(n_l - n_g)|\mathbb{A}|^{n_l})$. Given sufficient computational power, the optimal sequence of actions can be calculated with $n_l$ equal to the length of the episode, $n_g = 1$, and $n_m$ equal to the size of the state space.

In the original AML algorithm, the utility was maximized over single actions, i.e. $n_l = 1$ which also makes $n_g = 1$. The value of $n_m$ used in their experiments is not reported, however they show an example with $n_m = 6$. We run AML with all possible combination of values of $n_l \in \{1, 5, 10, 15\}$, $n_g \in \{1, n_l\}$ and $n_m = \{5, 10\}$ and define two versions: (1) Active Markov Localization (Fast): Generalized AML algorithm using the values of $n_l, n_g, n_m$ that maximize the performance while keeping the runtime comparable to ANL, and (2) Active Markov Localization (Slow): Generalized AML algorithm using the values of $n_l, n_g, n_m$ which maximize the performance while keeping the runtime for 1000 episodes below 24hrs (which is the training time of the proposed model) in each environment (See Appendix B for more details on the implementation of AML).

The perceptual model for both Markov Localization and Active Markov Localization needs to be specified separately. For the 2D environments, the perceptual model uses 1-hot vector representation of depth. For the 3D Environments, the perceptual model uses a pretrained Resnet-18 (He et al., 2016) model to calculate the feature representations for the agent observations and the memory images.

## 5.3 RESULTS

**2D Environments**    For the Maze2D environment, we run all models on mazes having size $7 \times 7$, $15 \times 15$ and $21 \times 21$ with varying episode lengths. We train all the models on randomly generated mazes and test on a fixed set of 1000 mazes (different from the mazes used in training). The results on the Maze2D environment are shown in Table 1. As seen in the table, the proposed model, Active Neural Localization, outperforms all the baselines on an average. The proposed method achieves a higher overall accuracy than AML (Slow) while being 100 times faster. Note that the runtime of AML for 1000 episodes is comparable to the total training time of the proposed model. The long runtime of AML (Slow) makes it infeasible to be used in real-time in many cases. When AML has comparable runtime, its performance drops by about 37% (AML (Fast)). We also observe that the difference in the performance of ANL and baselines is higher for smaller episode lengths. This indicates that ANL is more efficient (meaning it requires fewer actions to localize) in addition to being more accurate.

**3D Environments**    All the mazes in the Maze3D environment are of size $70\times70$ while the office environment environment is of size $70\times50$. The agent location is a continuous value in this range. Each cell roughly corresponds to an area of 40cm$\times$40cm in the real world. The set of memory images correspond to only about 6% of the total states. Likelihood of rest of the states are obtained by bilinear smoothing. All episodes have a fixed length of 30 actions. Although the size of the Office Map is $70\times50$, we represent Likelihood and Belief by a $70\times70$ in order to transfer the model between

**Table 2:** Results on the 3D environments. 'Time' refers to the number of seconds required to evaluate 1000 episodes with the corresponding method and 'Acc' stands for accuracy over 1000 episodes.

| Env | | Maze3D | | | | | | Unreal3D with lights | | All | Domain adaptation |
|---|---|---|---|---|---|---|---|---|---|---|---|
| Evaluation Setting | | Unseen Mazes Seen Textures | | | Unseen mazes Unseen textures | | | With lights | Without lights | | Maze3D to Unreal3D |
| No. of landmarks | | 10 | 5 | 0 | 10 | 5 | 0 | | | | |
| Markov Localization | Time | 2415 | 2470 | 2358 | 2580 | 2509 | 2489 | 2513 | 2541 | 2484.4 | - |
| (Resnet) | Acc | 0.716 | 0.657 | 0.641 | 0.702 | 0.669 | 0.652 | 0.517 | 0.249 | 0.600 | - |
| Active Markov | Time | 14231 | 12409 | 11662 | 15738 | 12098 | 11761 | 11878 | 5511 | 11911.0 | - |
| Localization (Fast) | Acc | 0.741 | 0.701 | 0.669 | 0.745 | 0.687 | 0.689 | 0.546 | 0.279 | 0.632 | - |
| Active Markov | Time | 48291 | 47424 | 43096 | 48910 | 44500 | 44234 | 47962 | 11205 | 41952.8 | - |
| Localization (Slow) | Acc | 0.759 | 0.749 | 0.694 | 0.787 | 0.730 | 0.720 | 0.577 | 0.302 | 0.665 | - |
| Active Neural | Time | 297 | 300 | 300 | 300 | 300 | 301 | 2750 | 2699 | 905.9 | 2756 |
| Localization | Acc | **0.889** | **0.859** | **0.852** | **0.858** | **0.839** | **0.871** | **0.934** | **0.505** | **0.826** | **0.921** |

both the 3D environments for domain adaptation. We also add a Gaussian noise of 5% standard deviation to all translations in 3D environments.

In the **Maze3D** environment, we vary the difficulty of the environment by varying the number of *landmarks* in the environment. Landmarks are defined to be walls with a unique texture. Each landmark is present only on a single wall in a single cell in the maze grid. All the other walls have a common texture making the map very ambiguous. We expect landmarks to make localization easier as the likelihood maps should have a lower entropy when the agent visits a landmark, which consequently should reduce the entropy of the Belief Map. We run experiments with 10, 5 and 0 landmarks. The textures of the landmarks are randomized during training. This technique of domain randomization has shown to be effective in generalizing to unknown maps within the simulation environment (Lample & Chaplot, 2016) and transferring from simulation to real-world (Tobin et al., 2017). In each experiment, the agent is trained on a set of 40 mazes and evaluated in two settings: (1) Unseen mazes with seen textures: the textures of each wall in the test set mazes have been seen in the training set, however the map design of the test set mazes are unseen and (2) Unseen mazes with unseen textures: both the textures and the map design are unseen. We test on a set of 20 mazes for each evaluation setting. Figure 3 shows examples for both the settings.

In the **Unreal3D** environment, we test the effectiveness of the model in adapting to dynamic lightning changes. We modified the the Office environment using the Unreal Game Engine Editor to create two scenarios: (1) Lights: where all the office lights are switched on; (2) NoLights: where all the office lights are switched off. Figure 3 shows sample agent observations with and without lights at the same locations. To test the model's ability to adapt to dynamic lighting changes, we train the model on the Office map with lights and test it on same map without lights. The memory images provided to the agent are taken while lights are switched on. Note that this is different as compared to the experiments on unseen mazes in Maze3D environment, where the agent is given memory images of the unseen environments.

The results for the 3D environments are shown in Table 2 and an example of the policy execution is shown in Figure 4[5]. The proposed model significantly outperforms all baseline models in all evaluation settings with the lowest runtime. We see similar trends of runtime and accuracy trade-off between the two version of AML as seen in the 2D results. The absolute performance of AML (Slow) is rather poor in the 3D environments as compared to Maze2D. This is likely due to the decrease in value of look-ahead parameter, $n_l$, to 3 and the increase in value of the greediness hyper-parameter, $n_g$ to 3, as compared to $n_l = 5, n_g = 1$ in Maze 2D, in order to ensure runtimes under 24hrs.

The ANL model performs better on the realistic Unreal environment as compared to Maze3D environment, as most scenes in the Unreal environment consists of unique landmarks while Maze3D environment consists of random mazes with same texture except those of the landmarks. In the Maze3D environment, the model is able to generalize well to not only unseen map design but also to unseen textures. However, the model doesn't generalize well to dynamic lighting changes in the Unreal3D environment. This highlights a current limitation of RGB image-based localization approaches as compared to depth-based approaches, as depth sensors are invariant to lighting changes.

---

[5] Demo videos: `https://devendrachaplot.github.io/projects/Neural-Localization`

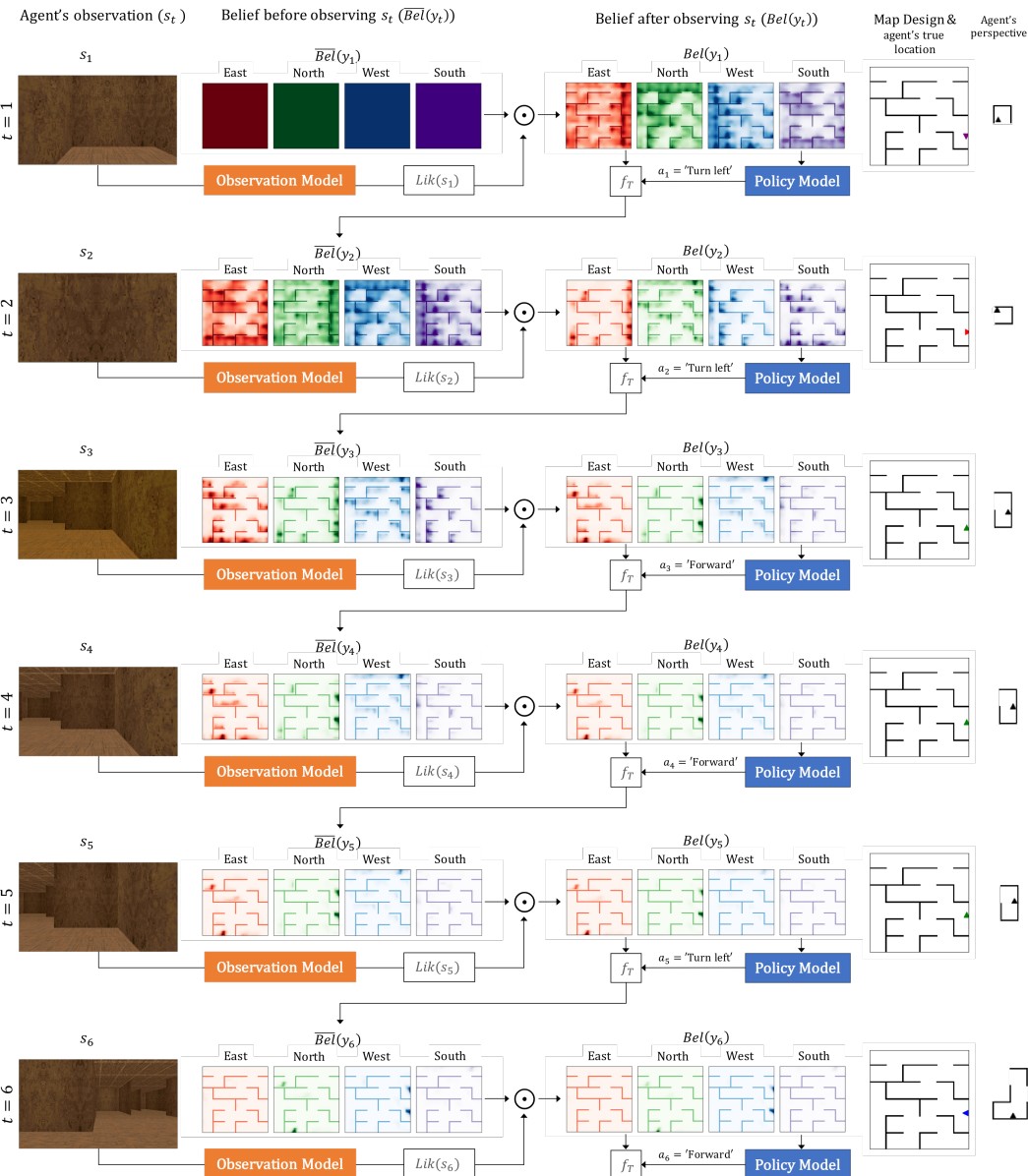

**Figure 4:** An example of the policy execution and belief propagation in the Maze3D Environment. The rows shows consecutive time steps in a episode. The columns show Agent's observation, the belief of its location before and after making the observation, the map design and agent's perspective of the world. Agent's true location is also marked in the map design (not visible to the agent). Belief maps show the probability of being at a particular location. Darker shades imply higher probability. The belief of its orientation and agent's true orientation are also highlighted by colors. For example, the Red belief map shows the probability of agent facing east direction at each x-y coordinate. Note that map design is not a part of the Belief Maps, it is overlayed on the Belief Maps for better visualization. At all time steps, all locations which look similar to agent's perspective have high probabilities in the belief map. The example shows the importance deciding actions while localizing. At $t = 3$, the agent is uncertain about its location as there are 4 positions with identical perspectives. The agent executes the optimal set of action to reduce its uncertainity, i.e. move forward and turn left, and successfully localizes.

**Domain Adaptation** We also test the ability of the proposed model to adapt between different simulation environments. The model trained on the Maze3D is directly tested on the Unreal3D Office Map without any fine-tuning. The results in Table 2 show that the model is able to generalize well to Unreal environment from the Doom Environment. We believe that the policy model generalizes well because the representation of belief and map design is common in all environments and policy model is based only on the belief and the map design, while the perceptual model generalizes well because it has learned to measure the similarity of the current image with the memory images as it was trained on environments with random textures. This property is similar to siamese networks used for one-shot image recognition (Koch et al., 2015).

## 6 CONCLUSION

In this paper, we proposed a fully-differentiable model for active global localization which uses structured components for Bayes filter-like belief propagation and learns a policy based on the belief to localize accurately and efficiently. This allows the policy and observation models to be trained jointly using reinforcement learning. We showed the effectiveness of the proposed model on a variety of challenging 2D and 3D environments including a realistic map in the Unreal environment. The results show that our model consistently outperforms the baseline models while being order of magnitudes faster. We also show that a model trained on random textures in the Doom simulation environment is able to generalize to photo-realistic Office map in the Unreal simulation environment. While this gives us hope that model can potentially be transferred to real-world environments, we leave that for future work. The limitation of the model to adapt to dynamic lightning can potentially be tackled by training the model with dynamic lightning in random mazes in the Doom environment. There can be several extensions to the proposed model too. The model can be combined with Neural Map (Parisotto & Salakhutdinov, 2017) to train an end-to-end model for a SLAM-type system and the architecture can also be utilized for end-to-end planning under uncertainity.

## ACKNOWLEDGEMENTS

This work was supported by Apple, IARPA DIVA award D17PC00340, and ONR award N000141512791. The authors would also like to thank Jian Zhang for helping with the Unreal environment and the NVidia for donating a DGX-1 deep learning machine and providing GPU support.

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

# A  BACKGROUND: REINFORCEMENT LEARNING

In the standard Reinforcement Learning Sutton & Barto (1998) setting, at each time step $t$, an agent receives a observation, $s_t$, from the environment, performs an action $a_t$ and receives a reward $r_t$. The goal is to learn a policy $\pi(a|s)$ which maximizes the expected return or the sum of discounted rewards $R_t = \Sigma_{t'=t}^{T} \gamma^{t'-t} r_{t'}$, where T is the time at which the episode terminates, and $\gamma \in [0,1]$ is a discount factor that determines the importance of future rewards.

Reinforcement learning methods can broadly be divided into value-based methods and policy-based methods. Policy-based methods parametrize the policy function which can be optimized directly to maximize the expected return ($\mathbb{E}[R_t]$) Sutton et al. (2000). While policy-based methods suffer from high variance, value-based methods typically use temporal difference learning which provides low variance estimates of the expected return. Actor-Critic methods (Barto et al., 1983; Sutton, 1984; Konda & Tsitsiklis, 1999) combine the benefits of both value-based methods by estimating both the value function, $V^\pi(s_t; \theta_v)$, as well as the policy function $\pi(a_t|s_t; \theta)$(Grondman et al., 2012).

REINFORCE family of algorithms (Williams, 1992) are popular for optimizing the policy function, which updates the policy parameters $\theta$ in the direction of $\nabla_\theta \log \pi(a_t|s_t; \theta) R_t$. Since this update is an unbiased estimate of $\nabla_\theta \mathbb{E}[R_t]$, its variance can be reduced by subtracting a baseline function, $b_t(s_t)$ from the expected return ($\nabla_\theta \log \pi(a_t|s_t; \theta)(R_t - b_t(s_t))$). When the estimate of the value function ($V^\pi(s_t)$) is used as the baseline, the resultant algorithm is called Advantage Actor-Critic, as the resultant policy gradient is scaled by the estimate of the *advantage* of the action $a_t$ in state $s_t$, $A(a_t, s_t) = Q(a_t, s_t) - V(s_t)$. The Asynchronous Advantage Actor-Critic algorithm (Mnih et al., 2016) uses a deep neural network to parametrize the policy and value functions and runs multiple parallel threads to update the network parameters.

In this paper, we use the A3C algorithm for all our experiments. We also use entropy regularization for improved exploration as described by (Mnih et al., 2016). In addition, we use the Generalized Advantage Estimator (Schulman et al., 2015) to reduce the variance of the policy gradient updates.

# B  IMPLEMENTATION DETAILS

## B.1  MODEL ARCHITECTURE DETAILS

The **perceptual model** for the 3D Environments receives RGB images of size 108x60. It consists of 2 Convolutional Layers. The first convolutional layer contains 32 filters of size 8x8 and stride of 4. The second convolutional layer contains 64 filters of size 4x4 with a stride of 2. The convolutional layers are followed by a fully-connected layer of size 512. The output of this fully-connected layer is used as the representation of the image while constructing the likelihood map. Figure 5 shows the architecture of the perceptual model in 3D environments. This architecture is adapted from previous work which is shown to perform well at playing deathmatches in Doom (Chaplot & Lample, 2017).

The **policy model** consists of two convolutional layers too. For the 2D environments, both the convolutional layers contain 16 filters of size 3 with stride of 1. For the 3D environments, the first convolutional layer contains 16 filters of size 7x7 with a stride of 3 and the second convolutional layer contains 16 filters of size 3x3 with a stride of 1. The convolutional layers are followed by a fully-connected layer of size 256. Figure 6 shows the architecture of the policy model in 3D environments.

We add action histroy of length 5 (last 5 actions) as well as the current time step as input to the policy model. We observed that action history input avoids the agent being stuck in alternating 'turn left' and 'turn right' actions whereas time step helps in accurately predicting the value function as the episode lengths are fixed in each environment. Each action in the action history as well as the current timestep are passed through an Embedding Layer to get an embedding of size 8. The embeddings of all actions and the time step are contacted with the 256-dimensional output of the fully-connected layer. The resultant vector is passed through two branches of single fully-connected layers to get the policy (actor layer with 3 outputs) and the value function (critic layer with 1 output).

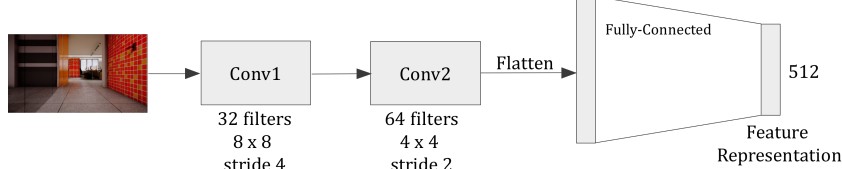

**Figure 5:** Figure showing the architecture of the perceptual model in 3D Environments.

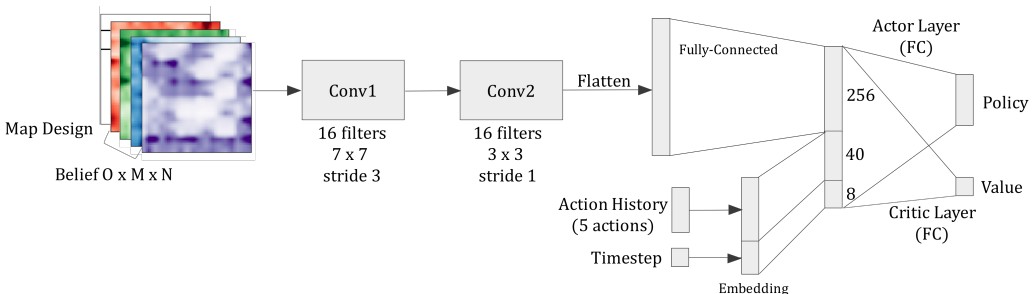

**Figure 6:** Figure showing the architecture of the policy model in 3D Environments.

### B.2 HYPER-PARAMETERS AND TRAINING DETAILS

All the models are trained with A3C using Stochastic Gradient Descent with a learning rate of 0.001. We use 8 threads for 2D experiments and 4 threads for 3D experiments. Each thread performed an A3C update after 20 steps. The weight for entropy regularization was 0.01. The discount factor ($\gamma$) for reinforcement learning was chosen to be 0.99. The gradients were clipped at 40. All models are trained for 24hrs of wall clock time. All the 2D experiments (including evaluation runtime benchmarks for baselines) were run on Intel(R) Xeon(R) CPU E5-2630 v4 @ 2.20GHz and all the 3D experiments were run on Intel(R) Core(TM) i7-6850K CPU @ 3.60GHz. While all the A3C training threads ran on CPUs, the Unreal engine also utilized a NVidia GeForce GTX 1080 GPU. The model with the best performance on the training environment is used for evaluation.

### B.3 TRANSITION FUNCTION

The transition function transforms the belief according to the action taken by the agent. For turn actions, the beliefs maps in each orientation are swapped according to the direction of the turn. For the move forward action, all probability values move one cell in the orientation of the agent, except those which are blocked by a wall (indicating a collision). Figure 7 shows sample outputs of the transition function given previous belief and action taken by the agent.

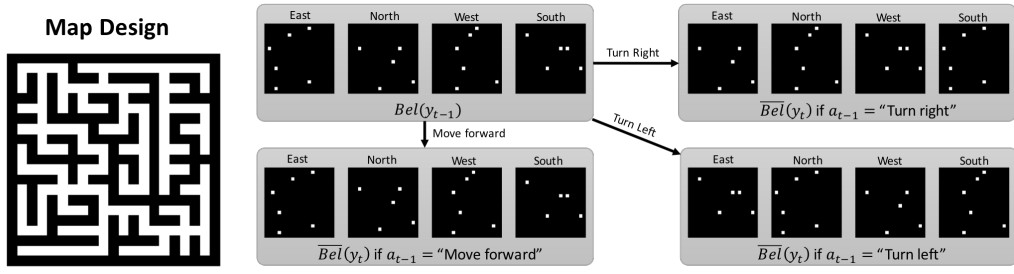

**Figure 7:** Sample output of the transition function ($f_T$) given previous belief and action taken by the agent. The map design is shown in the left.

### B.4 IMPLEMENTATION DETAILS OF ACTIVE MARKOV LOCALIZATION

In order to make our implementation of generalized AML as efficient as possible, we employ various techniques described by the authors, such as Pre-computation and Selective computation (Fox et al., 1998), along with other techniques such as hashing of expected entropies for action subsequences. The restrictions in runtime led to $n_l = 1, n_g = 1, n_m = 5$ in both 2D and 3D environments for AML (Fast), $n_l = 5, n_g = 1, n_m = 10$ in 2D environments for AML (Slow) and $n_l = 3, n_g = 3, n_m = 10$ in the 3D environments for AML (Slow).

The computation of expected entropies require the expected observation in the future states while rolling out a sequence of action. While it is possible to calculate these in 2D environments with depth-based observations, it is not possible to do this in 3D environments with RGB image observations. However, for comparison purposes we assume that AML has a perfect model of the environment and provide future observations by rolling out the action sequences in the simulation environment.

