# OpenReview forum: "Active Neural Localization"
_ICLR.cc/2018/Conference — Accept (Poster)_

### Official Review · AnonReviewer3 · 2017-11-27
**Convincing paper about an active learning neural version of Bayesian filter localisation**

**Rating:** 8
**Confidence:** 5

**Review:**

I have evaluated this paper for NIPS 2017 and gave it an "accept" rating at the time, but the paper was ultimately not accepted. This resubmission has been massively improved and definitely deserves to be published at ICLR.

This paper formulates the problem localisation on a known map using a belief network as an RL problem. The goal of the agent is to minimise the number of steps to localise itself (the agent needs to move around to accumulate evidence about its position), which corresponds to reducing the entropy of the joint distribution over a discretized grid over theta (4 orientations), x and y. The model is evaluated on a grid world, on textured 3D mazes with simplified motion (Doom environment) and on a photorealistic environment using the Unreal engine. Optimisation is done through A3C RL. Transfer from the crude simulated Doom environment to the photorealistic Unreal environment is achieved.

The belief network consists of an observation model, a motion prediction model that allows for translations along x or y and 90deg rotation, and an observation correction model that either perceives the depth in front of the agent (a bold and ambiguous choice) and matches it to the 2D map, or perceives the image in front of the agent. The map is part of the observation.

The algorithm outperforms Bayes filters for localisation in 2D and 3D and the idea of applying RL to minimise the entropy of position estimation is brilliant. Minor note: I am surprised that the cognitive map reference (Gupta et al, 2017) was dropped, as it seemed relevant.

---

> ### Author Response · Authors · 2017-12-23
> **Author response to AnonReviewer3**
>
> We thank the reviewer for their valuable comments and feedback.
>
> > Minor note: I am surprised that the cognitive map reference (Gupta et al, 2017) was dropped, as it seemed relevant.
> We agree that this reference is relevant, we have added the reference to the revision.

---

> > ### Comment · AnonReviewer3 · 2017-12-26
> > **Thank you for the correction**
> >
> > Having seen the other reviews and rebuttals, I maintain my rating at 8 (top 50%, clear accept).

---

### Official Review · AnonReviewer2 · 2017-11-28
**Provides Reasonable Evaluation but would Benefit from Clearer Motivation**

**Rating:** 6
**Confidence:** 4

**Review:**

The paper describes a neural network-based approach to active localization based upon RGB images. The framework employs Bayesian filtering to maintain an estimate of the agent's pose using a convolutional network model for the measurement (perception) function. A convolutional network models the policy that governs the action of the agent. The architecture is trained in an end-to-end manner via reinforcement learning. The architecture is evaluated in 2D and 3D simulated environments of varying complexity and compared favorably to traditional (structured) approaches to passive and active localization.

As the paper correctly points out, there is large body of work on map-based localization, but relatively little attention has been paid to decision theoretic formulations to localization, whereby the agent's actions are chosen in order to improve localization accuracy. More recent work instead focuses on the higher level objective of navigation, whereby any effort act in an effort to improve localization are secondary to the navigation objective. The idea of incorporating learned representations with a structured Bayesian filtering approach is interesting, but it's utility could be better motivated. What are the practical benefits to learning the measurement and policy model beyond (i) the temptation to apply neural networks to this problem and (ii) the ability to learn these in an end-to-end fashion? That's not to say that there aren't benefits, but rather that they aren't clearly demonstrated here. Further, the paper seems to assume (as noted below) that there is no measurement uncertainty and, with the exception of the 3D evaluations, no process noise.

The evaluation demonstrates that the proposed method yields estimates that are more accurate according to the proposed metric than the baseline methods, with a significant reduction in computational cost. However, the environments considered are rather small by today's standards and the baseline methods almost 20 years old. Further, the evaluation makes a number of simplifying assumptions, the largest being that the measurements are not subject to noise (the only noise that is present is in the motion for the 3D experiments). This assumption is clearly not valid in practice. Further, it is not clear from the evaluation whether the resulting distribution that is maintained is consistent (e.g., are the estimates over-/under-confident?). This has important implications if the system were to actually be used on a physical system. Further, while the computational requirements at test time are significantly lower than the baselines, the time required for training is likely very large. While this is less of an issue in simulation, it is important for physical deployments. Ideally, the paper would demonstrate performance when transferring a policy trained in simulation to a physical environment (e.g., using diversification, which has proven effective at simulation-to-real transfer).

Comments/Questions:

* The nature of the observation space is not clear.

* Recent related work has focused on learning neural policies for navigation, and any localization-specific actions are secondary to the objective of reaching the goal. It would be interesting to discuss how one would balance the advantages of choosing actions that improve localization with those in the context of a higher-level task (or at least including a cost on actions as with the baseline method of Fox et al.).

* The evaluation that assigns different textures to each wall is unrealistic.

* It is not clear why the space over which the belief is maintained flips as the robot turns and shifts as it moves.

* The 3D evaluation states that a 360 deg view is available. What happens when the agent can only see in one (forward) direction?

* AML includes a cost term in the objective. Did the author(s) experiment with setting this cost to zero?

* The 3D environments rely upon a particular belief size (70 x 70) being suitable for all environments. What would happen if the test environment was larger than those encountered in training?

* The comment that the PoseNet and VidLoc methods "lack a strainghtforward method to utilize past map data to do localization in a new environment" is unclear.

* The environments that are considered are quite small compared to the domains currently considered for

* Minor: It might be better to move Section 3 into Section 4 after introducing notation (to avoid redundancy).
* The paper should be proofread for grammatical errors (e.g., "bayesian" --> "Bayesian", "gaussian" --> "Gaussian")


UPDATES FOLLOWING AUTHORS' RESPONSE

(Apologies if this is a duplicate. I added a comment in light of the authors' response, but don't see it and so I am updating my review for completeness).

I appreciate the authors's response to the initial reviews and thank them for addressing several of my comments.

RE: Consistency
My concerns regarding consistency remain. For principled ways of evaluating the consistency of an estimator, see Bar-Shalom "Estimation with Applications to Tracking and Navigation".

RE: Measurement/Process Noise
The fact that the method assumes perfect measurements and, with the exception of the 3D experiments, no process noise is concerning as neither assumptions are valid for physical systems. Indeed, it is this noise in particular that makes localization (and its variants) challenging.

RE: Motivation
The response didn't address my comments about the lack motivation for the proposed method. Is it largely the temptation of applying an end-to-end neural method to a new problem? The paper should be updated to make the advantages over traditional approaches to active localization.

---

> ### Author Response · Authors · 2017-12-23
> **Author response to AnonReviewer2**
>
> We thank the reviewer for their valuable comments and feedback.
>
> > What are the practical benefits to learning the measurement and policy model?
> The example at the end of the paper (see Figure 4) highlights the importance of deciding actions for fast and accurate localization. We agree that the benefits can be better motivated in the introduction and we are looking into restructuring the paper to have a motivating example in the introduction.
>
> > it is not clear from the evaluation whether the resulting distribution that is maintained is consistent:
> Looking at the output of the model manually, it seems that the estimates are consistent. It is very difficult to quantify the consistency of the resulting distribution because there is no straightforward way to calculate the ground-truth distribution/posterior in 3D environments.
>
> > while the computational requirements at test time are significantly lower than the baselines, the time required for training is likely very large:
> We designed the Active Markov Localization (Slow) baseline keeping this in mind. The proposed model was trained for 24hrs for all experiments. AML (Slow) represents the Generalized AML algorithm using the values of hyperparameters which maximize the performance while keeping the runtime for 1000 episodes below 24hrs in each environment. This means the runtime of AML (Slow) is comparable to the training time of the proposed model. However, we agree that this point should be stated explicitly and we have made relevant changes in the paper.
>
> > The nature of the observation space is not clear.
> The observation space in 2D environments is just the depth of the one column in front of the agent and in 3D environments, it is the 108x60 RGB image showing the first-person view of the agent.
>
> > It is not clear why the space over which the belief is maintained flips as the robot turns and shifts as it moves.
> This happens due to the transition function as each channel represents a quantized orientation (North/East/West/South). The details of the transition function are provided in the appendix. For example, if the agent turns left, the probability of it facing north at any x-y coordinate becomes the probability of it facing west at the same-coordinate. This is why the belief flips when turning left. Similarly, the belief flips in the opposite direction when turning right and shifts when moving forward.
>
> > The 3D evaluation states that a 360 deg view is available. What happens when the agent can only see in one (forward) direction?
> This seems to be a misunderstanding. The agent only sees in one forward direction, it needs to take actions to turn around to get the view in other directions. This misunderstanding might be due to the likelihood and belief presented in 4 directions. Note that each of these 4 channels represents the likelihood/belief of the agent’s orientation being that direction, not the likelihood/belief of the view in that direction.
>
> > AML includes a cost term in the objective. Did the author(s) experiment with setting this cost to zero?
> In our environment, all actions have the same cost. This is equivalent to setting the cost to zero (i.e. it does not affect the optimal policy in the environment), but we found it helps the optimization of our model.
>
> > What would happen if the test environment was larger than those encountered in training?
> We will need to discretize the test environment such that its belief is at most the size of the training environment, i.e. 70x70. The discretization of the training environments can be changed according to the desired level of accuracy in the test environment. For example, if we discretize 35m x 35m environment to a grid of 35x35, each cell would be a length of 1m. Due to this discretization, the model can make errors up to 0.5m even if it predicts the correct cell. The discretization can be increased to 70x70 to reduce errors to 0.25m.
>
> > The comment that the PoseNet and VidLoc methods "lack a straightforward method to utilize past map data to do localization in a new environment" is unclear.
> The network weights in these models memorize the environment. The model has no way to ingest information about the map as input, thus the model trained in one map cannot be transferred to another map. These models need to be retrained on any new map.

---

### Official Review · AnonReviewer1 · 2017-11-28
**End-to-end training of two parts of an active localization system**

**Rating:** 7
**Confidence:** 4

**Review:**

This is an interesting paper that builds a parameterized network to select actions for a robot in a simulated environment, with the objective of quickly reaching an internal belief state that is predictive of the true state.  This is an interesting idea and it works much better than I would have expected.

In more careful examination it is clear that the authors have done a good job of designing a network that is partly pre-specified and partly free, in a way that makes the learning effective.  In particular
- the transition model is known and fixed (in the way it is used in the belief update process)
- the belief state representation is known and fixed (in the way it is used to decide whether the agent should be rewarded)
- the reward function is known and fixed (as above)
- the mechanics of belief update
But we learn
- the observation model
- the control policy

I'm not sure that global localization is still an open problem with known models.  Or, at least, it's not one of our worst.

Early work by Cassandra, Kurien, et al used POMDP models and solvers for active localization with known transition and observation models.   It was computationally slow but effective.

Similarly, although the online speed of your learned method is much better than for active Markov localization, the offline training cost is dramatically higher;  it's important to remember to be clear on this point.

It is not obvious to me that it is sensible to take the cosine similarity between the feature representation of the observation and the feature representation of the state to get the entry in the likelihood map.   It would be good to make it clear this is the right measure.

How is exploration done during the RL phase?  These domains are still not huge.

Please explain in more detail what the memory images are doing.

In general, the experiments seem to be well designed and well carried out, with several interesting extensions.

I have one more major concern:  it is not the job of a localizer to arrive at a belief state with high probability mass on the true state---it is the job of a localizer to have an accurate approximation of the true posterior under the prior and observations.   There are situations (in which, for example, the robot has gotten an unusual string of observations) in which it is correct for the robot to have more probability mass on a "wrong" state.  Or, it seems that this model may earn rewards for learning to make its beliefs overconfident.  It would be very interesting to see if you could find an objective that would actually cause the model to learn to compute the appropriate posterior.

In the end, I have trouble making a recommendation:
Con:  I'm not convinced that an end-to-end approach to this problem is the best one
Pro: It's actually a nice idea that seems to have worked out well
Con: I remain concerned that the objective is not the right one

My rating would really be something like 6.5 if that were possible.

---

> ### Author Response · Authors · 2017-12-23
> **Author response to AnonReviewer1**
>
> We thank the reviewer for their valuable comments and feedback.
>
> Concerns regarding the objective function:
> This is a very interesting point and we thank the reviewer for this observation. We agree that one of the tasks of the localizer is to accurately approximate the true posterior under the prior and observations. But another task is to learn to take actions which lead it to arrive at a belief state with high probability mass on the true location. We provide rewards only for the correct prediction of the location and not for the correct prediction of the posterior because of two primary reasons:
> - Defining an appropriate reward function for the true posterior would require some way of estimating the true posterior, which is very difficult especially in 3D environments.
> - We want the model to be penalized if it fails to take actions in order to reach a state where it can predict its correct location, even if its estimation of the posterior under the prior and the observations is accurate.
>
> The second point can potentially be mitigated by having an auxiliary loss on the belief, which back-propagates only through the perceptual model. This will only reward the perceptual model for predicting the true posterior, and the policy loss would still penalize the whole model for taking unfavorable actions. However, this will still require defining a reward or a loss function for the true posterior, which is difficult as there is generally no straightforward way of computing the ground-truth posterior in 3D environments with unknown models.
>
> > although the online speed of your learned method is much better than for active Markov localization, the offline training cost is dramatically higher:
> We designed the Active Markov Localization (Slow) baseline keeping this in mind. The proposed model was trained for 24hrs for all experiments. AML (Slow) represents the Generalized AML algorithm using the values of hyperparameters which maximize the performance while keeping the runtime for 1000 episodes below 24hrs in each environment. This means that the runtime of AML (Slow) is comparable to the training time of the proposed model. However, we agree that this point should be stated explicitly and we have made relevant changes in the paper.
>
> > How is exploration done during the RL phase?.
> Exploration is done implicitly using the stochastic policy in Asynchronous Advantage Actor-Critic method. As in the original A3C paper, to encourage exploration, we used an entropy loss scale of 0.01.
>
> > Please explain in more detail what the memory images are doing.
> Memory images are a part of the map information given to the agent in 3D Environments. They are used to calculate the likelihood given the current observation of the agent as follows: The perceptual model is used to get the feature representation of all the memory images and the current agent observation. The likelihood of each state in the set of memory images is calculated by taking the cosine similarity of the feature representation of the agent’s observation with the feature representation of the memory image.
>
> > It is not obvious to me that it is sensible to take the cosine similarity between the feature representation of the observation and the feature representation of the state to get the entry in the likelihood map:
> The basic assumption here is that images containing the same “landmark” (unique texture or object) would have similar representations (e.g. high inner product value). Taking cosine similarity is similar to the standard attention operation commonly used in Deep Learning, which is exponentiated inner product. The cosine similarity, in contrast, is scaled to remain within a range of values, which can help training stability and prevent the likelihood model from becoming too sharp.

---

### Public Comment · (anonymous) · 2018-03-10
**Link for code does not work?**

Could you made the code available -- the github link does not work? I am curious about the stability of the approach.

---

### Decision · Program_Chairs · 2018-01-29
**ICLR 2018 Conference Acceptance Decision**

**Decision:**

Accept (Poster)

**Comment:**

The paper proposes a neural net based method for active localization in a known map using a learnt perception model (convnet) and a learnt control policy combined with a set belief state representation. The method compares well to baselines and has good accuracy in 2d and 3d envs. All three reviewers are in favor of acceptance due to the novelty and competitive performance of the approach.